# MAXIMUM ENTROPY FLOW NETWORKS

**Gabriel Loaiza-Ganem**\*, **Yuanjun Gao**\* & **John P. Cunningham**
Department of Statistics
Columbia University
New York, NY 10027, USA
`{gl2480,yg2312,jpc2181}@columbia.edu`

## ABSTRACT

Maximum entropy modeling is a flexible and popular framework for formulating statistical models given partial knowledge. In this paper, rather than the traditional method of optimizing over the continuous density directly, we learn a smooth and invertible transformation that maps a simple distribution to the desired maximum entropy distribution. Doing so is nontrivial in that the objective being maximized (entropy) is a function of the density itself. By exploiting recent developments in normalizing flow networks, we cast the maximum entropy problem into a finite-dimensional constrained optimization, and solve the problem by combining stochastic optimization with the augmented Lagrangian method. Simulation results demonstrate the effectiveness of our method, and applications to finance and computer vision show the flexibility and accuracy of using maximum entropy flow networks.

## 1 INTRODUCTION

The maximum entropy (ME) principle (Jaynes, 1957) states that subject to some given prior knowledge, typically some given list of moment constraints, the distribution that makes minimal additional assumptions – and is therefore appropriate for a range of applications from hypothesis testing to price forecasting to texture synthesis – is that which has the largest entropy of any distribution obeying those constraints. First introduced in statistical mechanics by Jaynes (1957), and considered both celebrated and controversial, ME has been extensively applied in areas including natural language processing (Berger et al., 1996), ecology (Phillips et al., 2006), finance (Buchen & Kelly, 1996), computer vision (Zhu et al., 1998), and many more.

Continuous ME modeling problems typically include certain expectation constraints, and are usually solved by introducing Lagrange multipliers, which under typical assumptions yields an exponential family distribution (also called Gibbs distribution) with natural parameters such that the expectation constraints are obeyed. Unfortunately, fitting ME distributions in even modest dimensions poses significant challenges. First, optimizing the Lagrangian for a Gibbs distribution requires evaluating the normalizing constant, which is in general computationally very costly and error prone. Secondly, in all but the rarest cases, there is no way to draw samples independently and identically from this Gibbs distribution, even if one could derive it. Third, unlike in the discrete case where a number of recent and exciting works have addressed the problem of estimating entropy from discrete-valued data (Jiao et al., 2015; Valiant & Valiant, 2013), estimating differential entropy from data samples remains inefficient and typically biased. These shortcomings are critical and costly, given the common use of ME distributions for generating reference data samples for a null distribution of a test statistic. There is thus ample need for a method that can both solve the ME problem and produce a solution that is easy and fast to sample.

In this paper we develop maximum entropy flow networks (MEFN), a stochastic-optimization-based framework and algorithm for fitting continuous maximum entropy models. Two key steps are required. First, conceptually, we replace the idea of maximizing entropy over a density directly with maximizing, over the parameter space of an indexed function family, the entropy of the density induced by mapping a simple distribution (a Gaussian) through that optimized function. Modern

---

\*These authors contributed equally.

neural networks, particularly in variational inference (Kingma & Welling, 2013; Rezende & Mohamed, 2015), have successfully employed this same idea to generate complex distributions, and we look to similar technologies. Secondly, unlike most other objectives in this network literature, the entropy objective itself requires evaluation of the target density directly, which is unavailable in most traditional architectures. We overcome this potential issue by learning a smooth, invertible transformation that maps a simple distribution to an (approximate) ME distribution. Recent developments in normalizing flows (Rezende & Mohamed, 2015; Dinh et al., 2016) allow us to avoid biased and computationally inefficient estimators of differential entropy (such as the nearest-neighbor class of estimators like that of Kozachenko-Leonenko; see Berrett et al. (2016)). Our approach avoids calculation of normalizing constants by learning a map with an easy-to-compute Jacobian, yielding tractable probability density computation. The resulting transformation also allows us to reliably generate iid samples from the learned ME distribution. We demonstrate MEFN in detail in examples where we can access ground truth, and then we demonstrate further the ability of MEFN networks in equity option prices fitting and texture synthesis.

Primary contributions of this work include: *(i)* addressing the substantial need for methods to sample ME distributions; *(ii)* introducing ME problems, and the value of including entropy in a range of generative modeling problems, to the deep learning community; *(iii)* the novel use of *constrained* optimization for a deep learning application; and *(iv)* the application of MEFN to option pricing and texture synthesis, where in the latter we show significant increase in the diversity of synthesized textures (over current state of the art) by using MEFN.

## 2 BACKGROUND

### 2.1 MAXIMUM ENTROPY MODELING AND GIBBS DISTRIBUTION

We consider a continuous random variable $\mathbf{Z} \in \mathcal{Z} \subseteq \mathbb{R}^d$ with density $p$, where $p$ has differential entropy $H(p) = -\int p(\mathbf{z}) \log p(\mathbf{z}) d\mathbf{z}$ and support $\text{supp}(p)$. The goal of ME modeling is to find, and then be able to easily sample from, the maximum entropy distribution given a set of moment and support constraints, namely the solution to:

$$
\begin{aligned}
p^* \quad = \quad &\text{maximize} \quad H(p) \\
&\text{subject to} \quad E_{\mathbf{Z} \sim p}[T(\mathbf{Z})] = 0 \\
&\qquad\qquad\quad \text{supp}(p) = \mathcal{Z},
\end{aligned}
\tag{1}
$$

where $T(\mathbf{z}) = (T_1(\mathbf{z}), ..., T_m(\mathbf{z})) : \mathcal{Z} \to \mathbb{R}^m$ is the vector of known (assumed sufficient) statistics, and $\mathcal{Z}$ is the given support of the distribution. Under standard regularity conditions, the optimization problem can be solved by Lagrange multipliers, yielding an exponential family $p^*$ of the form:

$$
p^*(\mathbf{z}) \propto e^{\eta^\top T(\mathbf{z})} \mathbb{1}(\mathbf{z} \in \mathcal{Z})
\tag{2}
$$

where $\eta \in \mathbb{R}^m$ is the choice of natural parameters of $p^*$ such that $E_{p^*}[T(\mathbf{Z})] = 0$. Despite this simple form, these distributions are only in rare cases tractable from the standpoint of calculating $\eta$, calculating the normalizing constant of $p^*$, and sampling from the resulting distribution. There is extensive literature on finding $\eta$ numerically (Darroch & Ratcliff, 1972; Salakhutdinov et al., 2002; Della Pietra et al., 1997; Dudik et al., 2004; Malouf, 2002; Collins et al., 2002), but doing so requires computing normalizing constants, which poses a challenge even for problems with modest dimensions. Also, even if $\eta$ is correctly found, it is still not trivial to sample from $p^*$. Problem-specific sampling methods (such as importance sampling, MCMC, etc.) have to be designed and used, which is in general challenging (burn-in, mixing time, etc.) and computationally burdensome.

### 2.2 NORMALIZING FLOWS

Following Rezende & Mohamed (2015), we define a *normalizing flow* as the transformation of a probability density through a sequence of invertible mappings. Normalizing flows provide an elegant way of generating a complicated distribution while maintaining tractable density evaluation. Starting with a simple distribution $\mathbf{Z}_0 \in \mathbb{R}^d \sim p_0$ (usually taken to be a standard multivariate

Gaussian), and by applying $k$ invertible and smooth functions $f_i : \mathbb{R}^d \to \mathbb{R}^d (i = 1, ..., k)$, the resulting variable $\mathbf{Z}_k = f_k \circ f_{k-1} \circ \cdots \circ f_1(\mathbf{Z}_0)$ has density:

$$p_k(\mathbf{z}_k) = p_0(f_1^{-1} \circ f_2^{-1} \circ \cdots \circ f_k^{-1}(\mathbf{z}_k)) \prod_{i=1}^{k} |\det(J_i(\mathbf{z}_{i-1}))|^{-1}, \tag{3}$$

where $J_i$ is the Jacobian of $f_i$. If the determinant of $J_i$ can be easily computed, $p_k$ can be computed efficiently.

Rezende & Mohamed (2015) proposed two specific families of transformations for variational inference, namely planar flows and radial flows, respectively:

$$f_i(\mathbf{z}) = \mathbf{z} + \mathbf{u}_i h(\mathbf{w}_i^T \mathbf{z} + b_i) \qquad \text{and} \qquad f_i(\mathbf{z}) = \mathbf{z} + \beta_i h(\alpha_i, r_i)(\mathbf{z} - \mathbf{z}_i'), \tag{4}$$

where $b_i \in \mathbb{R}$, $\mathbf{u}_i, \mathbf{w}_i \in \mathbb{R}^d$ and $h$ is an activation function in the planar case, and where $\beta_i \in \mathbb{R}$, $\alpha_i > 0$, $\mathbf{z}_i' \in \mathbb{R}^d$, $h(\alpha, r) = 1/(\alpha + r)$ and $r_i = ||\mathbf{z} - \mathbf{z}_i'||$ in the radial. Recently Dinh et al. (2016) proposed a normalizing flow with convolutional, multiscale structure that is suitable for image modeling and has shown promise in density estimation for natural images.

## 3 MAXIMUM ENTROPY FLOW NETWORK (MEFN) ALGORITHM

### 3.1 FORMULATION

Instead of solving Equation 2, we propose solving Equation 1 directly by optimizing a transformation that maps a random variable $\mathbf{Z}_0$, with simple distribution $p_0$, to the ME distribution. Given a parametric family of normalizing flows $\mathcal{F} = \{f_\phi, \phi \in \mathbb{R}^q\}$, we denote $p_\phi(\mathbf{z}) = p_0(f_\phi^{-1}(\mathbf{z}))|\det(J_\phi(\mathbf{z}))|^{-1}$ as the distribution of the variable $f_\phi(\mathbf{Z}_0)$, where $J_\phi$ is the Jacobian of $f_\phi$. We then rewrite the ME problem as:

$$\phi^* = \text{maximize} \quad H(p_\phi) \tag{5}$$
$$\text{subject to} \quad E_{\mathbf{Z}_0 \sim p_0}[T(f_\phi(\mathbf{Z}_0))] = 0$$
$$\text{supp}(p_\phi) = \mathcal{Z}.$$

When $p_0$ is continuous and $\mathcal{F}$ is suitably general, the program in Equation 5 recovers the ME distribution $p_\phi$ exactly. With a flexible transformation family, the ME distribution can be well approximated. In experiments we found that taking $p_0$ to be a standard multivariate normal distribution achieves good empirical performance. Taking $p_0$ to be a bounded distribution (e.g. uniform distribution) is problematic for learning transformations near the boundary, and heavy tailed distributions (e.g. Cauchy distribution) caused similar trouble due to large numbers of outliers.

### 3.2 ALGORITHM

We solved Equation 5 using the augmented Lagrangian method. Denote $R(\phi) = E(T(f_\phi(\mathbf{Z}_0)))$, the augmented Lagrangian method uses the following objective:

$$L(\phi; \lambda, c) = -H(p_\phi) + \lambda^\top R(\phi) + \frac{c}{2}||R(\phi)||^2 \tag{6}$$

where $\lambda \in \mathbb{R}^m$ is the Lagrange multiplier and $c > 0$ is the penalty coefficient. We minimize Equation 6 for a non-decreasing sequence of $c$ and well-chosen $\lambda$. As a technical note, the augmented Lagrangian method is guaranteed to converge under some regularity conditions (Bertsekas, 2014). As is usual in neural networks, a proof of these conditions is challenging and not yet available, though intuitive arguments (see Appendix §A) suggest that most of them should hold. Due to the non rigorous nature of these arguments, we rely on the empirical results of the algorithm to claim that it is indeed solving the optimization problem.

For a fixed $(\lambda, c)$ pair, we optimize $L$ with stochastic gradient descent. Owing to our choice of network and the resulting ability to efficiently calculate the density $p_\phi(\mathbf{z}^{(i)})$ for any sample point

---

**Algorithm 1** Training the MEFN

1: initialize $\phi = \phi_0$, set $c_0 > 0$ and $\lambda_0$.
2: **for** Augmented Lagrangian iteration $k = 1, ..., k_{\max}$ **do**
3: **for** SGD iteration $i = 1, ..., i_{\max}$ **do**
4: Sample $\mathbf{z}^{(1)}, ..., \mathbf{z}^{(n)} \sim p_0$, get transformed variables $\mathbf{z}_\phi^{(i)} = f_\phi(\mathbf{z}^{(i)}), i = 1, ..., n$
5: Update $\phi$ by descending its stochastic gradient (using e.g. ADADELTA (Zeiler, 2012)):

$$\nabla_\phi L(\phi; \lambda_k, c_k) \approx \frac{1}{n} \sum_{i=1}^{n} \nabla_\phi \log p_\phi(\mathbf{z}_\phi^{(i)}) + \frac{1}{n} \sum_{i=1}^{n} \nabla_\phi T(\mathbf{z}_\phi^{(i)}) \lambda_k + c_k \frac{2}{n} \sum_{i=1}^{\frac{n}{2}} \nabla_\phi T(\mathbf{z}_\phi^{(i)}) \cdot \frac{2}{n} \sum_{i=\frac{n}{2}+1}^{n} T(\mathbf{z}_\phi^{(i)})$$

6: **end for**
7: Sample $\mathbf{z}^{(1)}, ..., \mathbf{z}^{(\tilde{n})} \sim p_0$, get transformed variables $\mathbf{z}_\phi^{(i)} = f_\phi(\mathbf{z}^{(i)}), i = 1, ..., \tilde{n}$
8: Update $\lambda_{k+1} = \lambda_k + c_k \frac{1}{\tilde{n}} \sum_{i=1}^{\tilde{n}} T(\mathbf{z}_\phi^{(i)})$
9: Update $c_{k+1} \geq c_k$ (see text for detail)
10: **end for**

---

$\mathbf{z}^{(i)}$ (which are easy-to-sample iid draws from the multivariate normal $p_0$), we compute the unbiased estimator of $H(p_\phi)$ with:

$$H(p_\phi) \approx -\frac{1}{n} \sum_{i=1}^{n} \log p_\phi(f_\phi(\mathbf{z}^{(i)})) \tag{7}$$

$R(\phi)$ can also be estimated without bias by taking a sample average of $\mathbf{z}^{(i)}$ draws. The resulting optimization procedure is detailed in Algorithm 1, of which step 9 requires some detail: denoting $\phi_k$ as the resulting $\phi$ after $i_{max}$ SGD iterations at the augmented Lagrangian iteration $k$, the usual update rule for $c$ (Bertsekas, 2014) is:

$$c_{k+1} = \begin{cases} \beta c_k, \text{if } ||R(\phi_{k+1})|| > \gamma ||R(\phi_k)|| \\ c_k, \text{otherwise} \end{cases} \tag{8}$$

where $\gamma \in (0, 1)$ and $\beta > 1$. Monte Carlo estimation of $R(\phi)$ sometimes caused $c$ to be updated too fast, causing numerical issues. Accordingly, we changed the hard update rule for $c$ to a probabilistic update rule: a hypothesis test is carried out with null hypothesis $H_0 : E[||R(\phi_{k+1})||] = E[\gamma ||R(\phi_k)||]$ and alternative hypothesis $H_1 : E[||R(\phi_{k+1})||] > E[\gamma ||R(\phi_k)||]$. The $p$-value $p$ is computed, and $c_{k+1}$ is updated to $\beta c_k$ with probability $1 - p$. We used a two-sample $t$-test to calculate the $p$-value. What results is a robust and novel algorithm for estimating maximum entropy distributions, while preserving the critical properties of being both easy to calculate densities of particular points, and being trivially able to produce truly iid samples.

## 4 EXPERIMENTS

We first construct an ME problem with a known solution (§4.1), and we analyze the MEFN algorithm with respect to the ground truth and to an approximate Gibbs solution. These examples test the validity of our algorithm and illustrate its performance. §B and §4.3 then applies the MEFN to a financial data application (predicting equity option values) and texture synthesis, respectively, to illustrate the flexibility and practicality of our algorithm.

For §4.1 and §B, We use 10 layers of planar flow with a final transformation $g$ (specified below) that transforms samples to the specified support, and use with ADADELTA (Zeiler, 2012). For §4.3 we use real NVP structure and use ADAM (Kingma & Ba, 2014) with learning rate = 0.001. For all our experiments, we use $i_{max} = 3000$, $\beta = 4$, $\gamma = 0.25$. For §4.1 and §B we use $n = 300$, $\tilde{n} = 1000$, $k_{\max} = 10$; For §4.3 we use $n = \tilde{n} = 2$, $k_{\max} = 8$.

### 4.1 A MAXIMUM ENTROPY PROBLEM WITH KNOWN SOLUTION

Following the setup of the typical ME problem, suppose we are given a specified support $\mathcal{S} = \{\mathbf{z} = (z_1, \ldots, z_{d-1}) : z_i \geq 0 \text{ and } \sum_{k=1}^{d-1} z_k \leq 1\}$ and a set of constraints $E[\log Z_k] = \kappa_k (k = 1, ..., d)$,

where $Z_d = 1 - \sum_{k=1}^{d-1} Z_k$. We then write the maximum entropy program:

$$
\begin{aligned}
p^* \quad = \quad & \text{maximize} \quad H(p) && (9) \\
& \text{subject to} \quad E_{\mathbf{Z} \sim p}[\log Z_k - \kappa_k] = 0 \quad \forall k = 1, ..., d \\
& \text{supp}(p) = \mathcal{S}.
\end{aligned}
$$

This is a general ME problem that can be solved via the MEFN. Of course, we have particularly chosen this example because, though it may not obviously appear so, the solution has a standard and tractable form, namely the Dirichlet. This choice allows us to consider a complicated optimization program that happens to have known global optimum, providing a solid test bed for the MEFN (and for the Gibbs approach against which we will compare). Specifically, given a parameter $\alpha \in \mathbb{R}^d$, the Dirichlet has density:

$$
p(z_1, \ldots, z_{d-1}) = \frac{1}{B(\alpha)} \prod_{k=1}^{d} z_k^{\alpha_k - 1} \mathbb{1}\left((z_1, \ldots, z_{d-1}) \in \mathcal{S}\right) \tag{10}
$$

where $B(\alpha)$ is the multivariate Beta function, and $z_d = 1 - \sum_{k=1}^{d-1} z_k$. Note that this Dirichlet is a distribution on $\mathcal{S}$ and not on the $(d-1)$-dimensional simplex $\mathcal{S}^{d-1} = \{(z_1, \ldots, z_d) : z_k \geq 0 \text{ and } \sum_{k=1}^{d} z_k = 1\}$ (an often ignored and seemingly unimportant technicality that needs to be correct here to ensure the proper transformation of measure). Connecting this familiar distribution to the ME problem above, we simply have to choose $\alpha$ such that $\kappa_k = \psi(\alpha_k) - \psi(\alpha_0)$ for $k = 1, ..., d$, where $\alpha_0 = \sum_{k=1}^{d} \alpha_k$ and $\psi$ is the digamma function. We then can pose the above ME problem to the MEFN and compare performance against ground truth. Before doing so, we must stipulate the transformation $g$ that maps the Euclidean space of the multivariate normal $p_0$ to the desired support $\mathcal{S}$. Any sensible choice will work well (another point of flexibility for the MEFN); we use the standard transformation:

$$
g(z_1, ..., z_{d-1}) = \left( \frac{e^{z_1}}{\sum_{k=1}^{d-1} e^{z_k} + 1}, ..., \frac{e^{z_{d-1}}}{\sum_{k=1}^{d-1} e^{z_k} + 1} \right)^{\top} \tag{11}
$$

Note that the MEFN outputs vectors in $\mathbb{R}^{d-1}$, and not $\mathbb{R}^d$, because the Dirichlet is specified as a distribution on $\mathcal{S}$ (and not on the simplex $\mathcal{S}^{d-1}$). Accordingly, the Jacobian is a square matrix and its determinant can be computed efficiently using the matrix determinant lemma. Here, $p_0$ is set to the $(d-1)$-dimensional standard normal.

We proceed as follows: We choose $\alpha$ and compute the constraints $\kappa_1, ..., \kappa_d$. We run MEFN pretending we do not know $\alpha$ or the Dirichlet form. We then take a random sample from the fitted distribution and a random sample from the Dirichlet with parameter $\alpha$, and compare the two samples using the maximum mean discrepancy (MMD) kernel two sample test (Gretton et al., 2012), which assesses the fit quality. We take the sample size to be 300 for the two sample kernel test. Figure 1 shows an example of the transformation from normal (left panel) to MEFN (middle panel), and comparing that to the ground truth Dirichlet (right panel). The MEFN and ground truth Dirichlet densities shown in purple match closely, and the samples drawn (red) indeed appear to be iid draws from the same (maximum entropy) distribution in both cases.

Additionally, the middle panel of Figure 1 shows an important cautionary tale that foreshadows our texture synthesis results (§4.3). One might suppose that satisfying the moment matching constraints is adequate to produce a distribution which, if not technically the ME distribution, is still interestingly variable. The middle panel shows the failure of this intuition: in dark green, we show a network trained to simply match the moments specified above, and the resulting distribution quite poorly expresses the variability available to a distribution with these constraints, leading to samples that are needlessly similar. Given the substantial interest in using networks to learn implicit generative models (e.g., Mohamed & Lakshminarayanan (2016)), this concern is particularly relevant and highlights the importance of considering entropy.

Figure 2 quantitatively analyzes these results. In the left panel, for a specific choice of $\alpha = (1, 2, 3)$, we show our unbiased entropy estimate of the MEFN distribution $p_\phi$ as a function of the number of SGD iterations (red), along with the ground truth maximum entropy $H(p^*)$ (green line). Note

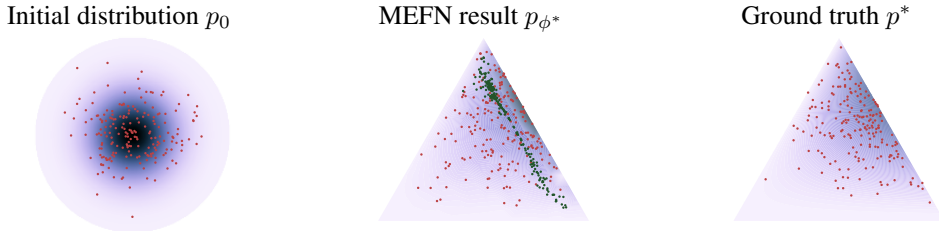

Figure 1: Example results from the ME problem with known Dirichlet ground truth. *Left panel*: The normal density $p_0$ (purple) and iid samples from $p_0$ (red points). *Middle panel*: The MEFN transforms $p_0$ to the desired maximum entropy distribution $p_{\phi^*}$ on the simplex (calculated density $p_{\phi^*}$ in purple). Truly iid samples are easily drawn from $p_{\phi^*}$ (red points) by drawing from $p_0$ and mapping those points through $f_{\phi^*}$. Shown in the middle panel are the same points in the top left panel mapped through $f_{\phi^*}$. Samples corresponding to training the same network as MEFN to simply match the specified moments (ignoring entropy) are also shown (dark green points; see text). *Right panel*: The ground truth (in this example, known to be Dirichlet) distribution in purple, and iid samples from it in red.

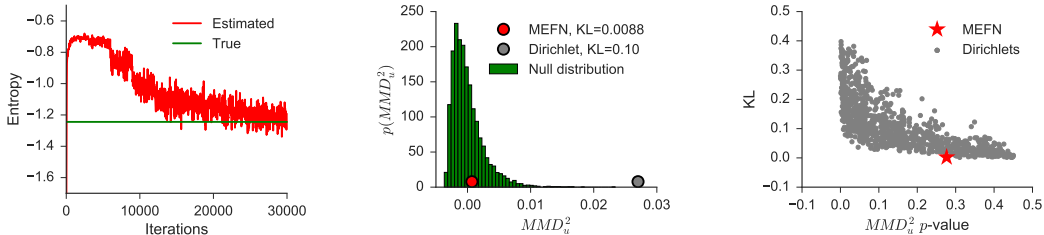

Figure 2: Quantitative analysis of simulation results. See text for description.

that the MEFN stabilizes at the correct value (as a stochastic estimator, variance around that value is expected). In the middle panel, we show the distribution of MMD values for the kernel two sample test, as well as the observed statistic for the MEFN (red) and for a randomly chosen Dirichlet distribution (gray; chosen to be close to the true optimum, making a conservative comparison). The MMD test does not reject MEFN as being different from the true ME distribution $p^*$, but it does reject a Dirichlet whose $KL$ to the true $p^*$ is small (see legend). In the right panel, for many different Dirichlets in a small grid around a single true $p^*$, the kernel two sample test statistic is computed, the MMD $p$-value is calculated, as is the $KL$ to the true distribution. We plot a scatter of these points in grey, and we plot the particular MEFN solution as a red star. We see that for other Dirichlets with similar $KL$ to the true distribution as the MEFN distribution, the $p$-values seem uniform, meaning that the $KL$ to the true is indeed very small. Again this is conservative, as the grey points have access to the known Dirichlet form, whereas the MEFN considered the entire space (within its network capacity) of $\mathcal{S}$ supported distributions. Given this fact, the performance of MEFN is impressive.

## 4.2 RISK-NEUTRAL ASSET PRICING

We illustrate the flexibility and practicality of our algorithm extracting the risk-neutral asset price probability based on option prices, an active and interesting area for ME models. We find that MEFN and the classic Gibbs approach yield comparable performances. Owing to space limitations we have placed these results in Appendix §B.

## 4.3 MODELING IMAGES OF TEXTURES

Constructing generative models to generate random images with certain texture structure is an important task in computer vision. A line of texture synthesis research proceeds by first extracting a set

of features that characterizes the target texture and then generate images that match the features. The seminal work of Zhu et al. (1998) proposes constructing texture models under the ME framework, where features (or filters) of the given texture image are adaptively added in the model and a Gibbs distribution whose expected feature matches the target texture is learnt. One major difficulty with the method is that both model learning and image generation involve sampling from a complicated Gibbs distribution. More recent works exploit more complicated features (Portilla & Simoncelli, 2000; Gatys et al., 2015; Ulyanov et al., 2016). Ulyanov et al. (2016) propose the *texture net*, which uses a texture loss function by using the Gram matrices of the outputs of some convolutional layers of a pre-trained deep neural network for object recognition.

While the use of these complicated features does provide high-quality synthetic texture images, that work focuses exclusively on generating images that match these feature (moments). Importantly, this network focuses only on generating feature-matching images without using the ME framework to promote the diversity of the samples. Doing so can be deeply problematic: in Figure 1 (middle panel), we showed the lack of diversity resulting from only moment matching in that Dirichlet setting, and further we note that the extreme pathology would result in a point mass on the training image – a global optimum for this objective, but obviously a terrible generative model for synthesizing textures. Ideally, the MEFN will match the moments *and* promote sample diversity.

We applied MEFN to texture synthesis with an RGB representation of the $224 \times 224$ pixel images , $\mathbf{z} \in \mathcal{Z} = [0,1]^d$, where $d = 224 \times 224 \times 3$. We follow Ulyanov et al. (2016) (we adapted `https://github.com/ProofByConstruction/texture-networks`) to create a texture loss measure $T(\mathbf{z}) : [0,1]^d \to \mathbb{R}$, and aim to sample a diverse set of images with small moment violation. For the transformation family $\mathcal{F}$ we use the real NVP network structure proposed in Dinh et al. (2016) (we adapted `https://github.com/taesung89/real-nvp`). We use 3 residual blocks with 32 feature maps for each coupling layer and downscale 3 times. For fair comparison, we use the same real NVP structure for both[1], implemented in TensorFlow (Abadi et al., 2016).

As is shown in top row of figure 3, both methods generate visually pleasing images capturing the texture structure well. The bottom row of Figure 3 shows that texture cost (left panel) is similar for both methods, while MEFN generates figures with much larger entropy than the texture network formulation (middle panel), which is desirable (as previously discussed). The bottom right panel of figure 3 compares the marginal distribution of the RGB values sampled from the networks: we found that MEFN generates a more variable distribution of RGB values than the texture net. Further results are in Appendix §C.

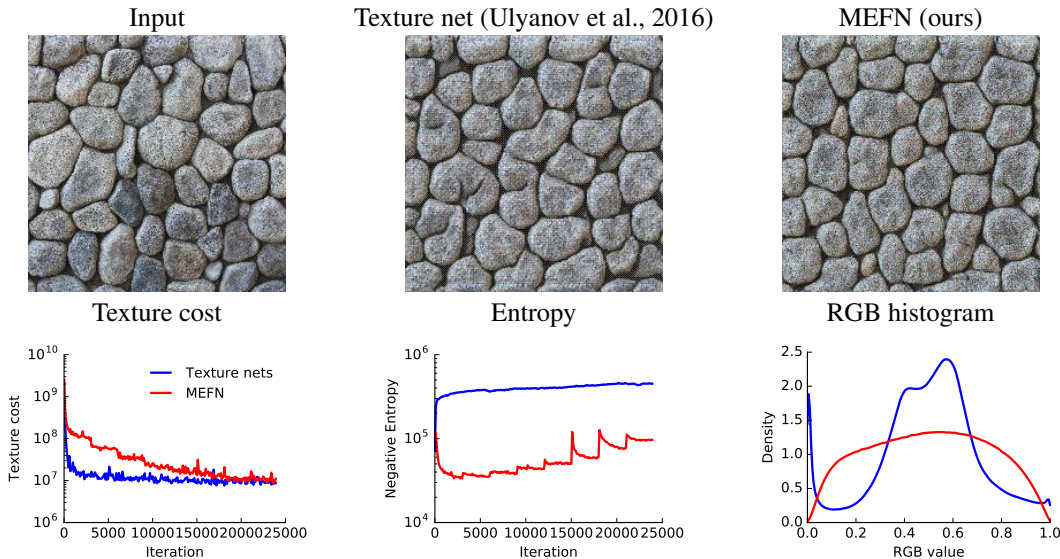

Figure 3: Analysis of texture synthesis experiment. See text for description.

---

[1]Ulyanov et al. (2016) use a quite different generative network structure, which is not invertible and is therefore infeasible for entropy evaluation, so we replace their generative network by the real NVP structure.

We compute in Table 1 the average pairwise Euclidean distance between randomly sampled images ($d_{L^2} = \text{mean}_{i \neq j} \|\mathbf{z}_i - \mathbf{z}_j\|_2^2$), and MEFN gives higher $d_{L^2}$, quantifying diversity across images. We also consider an ANOVA-style analysis to measure the diversity of the images, where we think of the RGB values for the same pixel across multiple images as a group, and compute the within and between group variance. Specifically, denoting $z_i^k$ as the pixel value for a specific pixel $k = 1, ..., d$ for an image $i = 1, ...., n$. We partition the total sum of square $\text{SST} = \sum_{i,k}(z_i^k - \bar{z})^2$ as the within group error $\text{SSW} = \sum_{i,k}(z_i^k - \bar{z}^k)^2$ and between group error $\text{SSB} = \sum_{i,k} n(\bar{z}^k - \bar{z})^2$, where $\bar{z}$ and $\bar{z}^k$ are the mean pixel values across all data and for a specific pixel $k$. Ideally we want the samples to exhibit large variability across images (large SSW, within a group/pixel) and no structure in the mean image (small SSB, across groups/pixels). Indeed, the MEFN has a larger SSW, implying higher variability around the mean image, a smaller SSB, implying the stationarity of the generated samples, and a larger SST, implying larger total variability also. The MEFN produces images that are conclusively more variable without sacrificing the quality of the texture, implicating the broad utility of ME.

Table 1: Quantitative measure of image diversity using 20 randomly sampled images

| Method | $d_{L^2}$ | SST | SSW | SSB |
|---|---|---|---|---|
| Texture net | 11534 | 128680 | 109577 | 19103 |
| MEFN | 17014 | 175604 | 161639 | 13964 |

## 5 CONCLUSION

In this paper we propose a general framework for fitting ME models. This approach is novel and has three key features. First, by learning a transformation of a simple distribution rather than the distribution itself, we are able to avoid explicitly computing an intractable normalizing constant for the ME distribution. Second, by combining stochastic optimization with the augmented Lagrangian method, we can fit the model efficiently, allowing us to evaluate the ME density of any point simply and accurately. Third, critically, this construction allows us to trivially sample iid from a ME distribution, extending the utility and efficiency of the ME framework more generally. Also, accuracy equivalent to the classic Gibbs approach is in itself a contribution (owing to these other features). We illustrate the MEFN in both a simulated case with known ground truth and real data examples.

There are a few recent works encouraging sample diversity in the setting of texture modeling. Ulyanov et al. (2017) extended Ulyanov et al. (2016) by adding a penalty term using the Kozachenko-Leonenko estimator Kozachenko & Leonenko (1987) of entropy. Their generative network is an arbitrary deep neural network rather than a normalizing flow, which is more flexible but cannot give the probability density of each sample easily so as to compute an unbiased estimator of the entropy. Kozachenko-Leonenko is a biased estimator for entropy and requires a fairly large number of samples to get good performance in high-dimensional settings, hindering the scalability and accuracy of the method; indeed, our choice of normalizing flow networks was driven by these practical issues with Kozachenko-Leonenko. Lu et al. (2016) extended Zhu et al. (1998) by using a more flexible set of filters derived from a pre-trained deep neural networks, and using parallel MCMC chains to learn and sample from the Gibbs distribution. Running parallel MCMC chains results in diverse samples but can be computationally intensive for generating each new sample image. Our MEFN framework enables truly iid sampling with the ease of a feed forward network.

ACKNOWLEDGMENTS

We thank Evan Archer for normalizing flow code, and Xuexin Wei, Christian Andersson Naesseth and Scott Linderman for helpful discussion. This work was supported by a Sloan Fellowship and a McKnight Fellowship (JPC).

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

## A    Augmented Lagrangian conditions

We give a more thorough discussion of the regularity conditions which ensure that the Augmented Lagrangian method will work. The goal of this section is simply to state these conditions and give intuitive arguments about why some should hold in our case, not to attempt to prove that they indeed hold. The conditions (Bertsekas, 2014) are:

- There exists a strict local minimum $\phi^*$ of the optimization problem of Equation 5:

  If the function class $\mathcal{F}$ is rich enough that it contains a true solver of the maximum entropy problem, then a global optimum exists. Although not rigorous, we would expect that even in the finite expressivity case that a global optimum remains, and indeed, recent theoretical work (Raghu et al., 2016; Poole et al., 2016) has gotten close to proving this.

- $\phi^*$ is a regular point of the optimization problem, that is, the rows of $\nabla_\phi R(\phi^*)$ are linearly independent:

  Again, this is not formal, but we should not expect this to cause any issues. This clearly depends on the specific form of $T$, but the condition basically says that there should not be redundant constraints at the optimum, so if $T$ is reasonable this shouldn't happen.

- $H(p_\phi)$ and $R(\phi)$ are twice continuously differentiable on a neighborhood around $\phi^*$:

  This holds by the smoothness of the normalizing flows.

- $y^\top \nabla_\phi^2 L(\phi^*; \lambda^*, 0)y > 0$ for every $y \neq 0$ such that $\nabla_\phi R(\phi^*)y = 0$, where $\lambda^*$ is the true Lagrange multiplier:

  This condition is harder to justify. It would appear it is just asking that the Lagrangian (not the augmented Lagrangian) be strictly convex in feasible directions, but it is actually stronger than this and some simple functions might not satisfy the property. For example, if the function we are optimizing was $x^4$ and we had no constraints, the Lagrangian's Hessian would be $12x^2$, which is $0$ at $x^* = 0$ thus not satisfying the condition. Importantly, these conditions are sufficient but not necessary, so even if this doesn't hold the augmented Lagrangian method might work (it certainly would for $x^4$). Because of this and the non-rigorous justifications of the first two conditions, we left these conditions for the appendix and relied instead on the empirical performance to justify that we are indeed recovering the maximum entropy distribution.

If all of these conditions hold, the augmented Lagrangian (for large enough $c$ and $\lambda$ close enough to $\lambda^*$) has a unique optimum in a neighborhood around $\phi^*$ that is close to $\phi^*$ (as $\lambda \to \lambda^*$ it converges to $\phi^*$) and its hessian at this optimum is positive-definite. Furthermore, $\lambda_k \to \lambda$. This implies that gradient descent (with the usual caveats of being started close enough to the solution and with the right steps) will correctly recover $\phi^*$ using the augmented Lagrangian method. This of course just guarantees convergence to a local optimum, but if there are no additional assumptions such as convexity, it can be very hard to ensure that it is indeed a global optimum. Some recent research has attempted to explain why optimization algorithms perform so well for neural networks (Choromanska et al., 2015; Kawaguchi, 2016), but we leave such attempts for our case for future research.

## B    Risk-neutral asset price

We extract the risk-neutral asset price probability distribution based on option prices, an active and interesting area for ME models. We give a brief introduction of the problem and refer interested readers to see Buchen & Kelly (1996) for a more detailed explanation. Denoting $S_t$ as the price of an asset at time $t$, the buyer of a European call option for the stock that expires at time $t_e$ with strike price $K$ will receive a payoff of $c_K = (S_{t_e} - K)_+ = \max(S_{t_e} - K, 0)$ at time $t_e$. Under the efficient market assumption, the risk-neutral probability distribution for the stock price at time $t_e$ satisfies:

$$c_K = D(t_e)E_q[(S_{t_e} - K)_+], \tag{12}$$

where $D(t_e)$ is the risk-free discount factor and $q$ is the risk-neutral measure. We also have that, under the risk-neutral measure, the current stock price $S_0$ is the discounted expected value of $S_{t_e}$:

$$S_0 = D(t_e)E_q(S_{t_e}). \tag{13}$$

When given $m$ options that expire at time $t_e$ with strikes $K_1, ..., K_m$ and prices $c_{K_1}, ..., c_{K_m}$, we get $m$ expectation constraints on $q(S_{t_e})$ from Equation 12, together with Equation 13, we have $m + 1$ expectation constraints in total. With that partial knowledge we can approximate $q(S_{t_e})$, which is helpful for understanding the market expected volatility and identify mispricing in option markets, etc.

Inferring the risk-neutral density of asset price from a finite number of option prices is an important question in finance and has been studied extensively (Buchen & Kelly, 1996; Borwein et al., 2003; Bondarenko, 2003; Figlewski, 2008). One popular method proposed by Buchen & Kelly (1996) estimates the probability density as the maximum entropy distribution satisfying the expectation constraints and a positivity support constraint by fitting a Gibbs distribution, which results in a piece-wise linear log density:

$$p(z) \propto \exp \left\{ \eta_0 z + \sum_{i=1}^{m} \eta_i (z - K_i)_+ \right\} \mathbb{1} \left( z \geq 0 \right) \tag{14}$$

and optimize the distribution with numerical methods. Here we compare the performance of the MEFN algorithm with the method proposed in Buchen & Kelly (1996). To enforce the positivity constraint we choose $g(z) = e^{az+b}$, where $a$ and $b$ are additional parameters.

We collect the closing price of European call options on Nov. 1 2016 for the stock AAPL (Apple inc.) that expires on $t_e =$ Jun. 16 2017. We use $m = 4$ of the options with highest trading volume as training data and the rest as testing data. On the left panel of figure 4, we show the fitted risk-neutral density of $S_{t_e}$ by MEFN (red line) with that of the fitted Gibbs distribution result (blue line). We find that while the distributions share similar location and variability, the distribution inferred by MEFN is smoother and arguably more plausible. In the middle panel we show a Q-Q plot of the quantiles of the MEFN and Gibbs distributions. We can see that the quantile pairs match the identity closely, which should happen if both methods recovered the exact same distribution. This highlights the effectiveness of MEFN. There does exist a small mismatch in the tails: the distribution inferred by MEFN has slightly heavier tails. This mismatch is difficult to interpret: given that both the Gibbs and MEFN distributions are fit with option price data (and given that one can observe at most one value from the distribution, namely the stock price at expiration), it is fundamentally unclear which distribution is superior, in the sense of better capturing the true ME distribution's tails. On the right panel we show the fitted option price for the two fitted distributions (for each strike price, we can recover the fitted option price by Equation 12). We noted that the fitted option price and strike price lines for both methods are very similar (they are mostly indiscernible on the right panel of figure 4). We also compare the fitted performance on the test data by computing the root mean square error for the fitted and test data. We observe that the predictive performances for both methods are comparable.

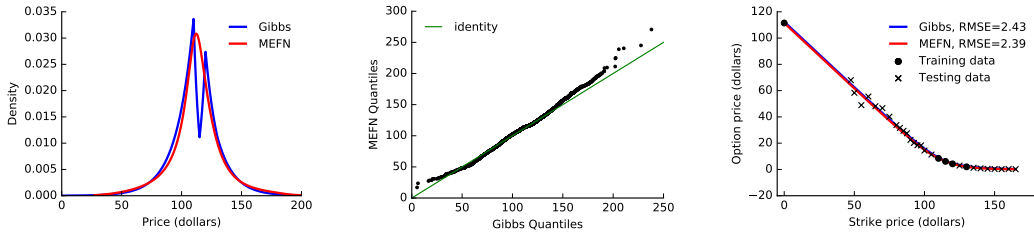

Figure 4: Constructing risk-neutral measure from observed option price. *Left panel*: fitted risk-neutral measure by Gibbs and MEFN method. *Middle panel*: Q-Q plot for the quantiles from the distributions on the left panel. *Right panel*: observed and fitted option price for different strikes.

We note that for this specific application, there are practical concerns such as the microstructure noise in the data and inefficiency in the market, etc. Applying a pre-processing procedure and incorporating prior assumptions can be helpful for getting a more full-fledged method (see e.g. Figlewski (2008)). Here we mainly focus on illustrating the ability of the MEFN method to approximate the ME distribution for non-typical distributions. Future work for this application includes fitting a risk-neutral distribution for multi-dimensional assets by incorporating dependence structure on assets.

## C    MODELING IMAGES OF TEXTURES

We tried our texture modeling approach with many different textures, and although MEFN samples don't always exhibit more visual diversity than samples obtained from the texture network, they always have more entropy as in figure 3. Figure 5 shows two positive examples, i.e. textures in which samples from MEFN do exhibit higher visual diversity than those from the texture network, as well as a negative example, in which MEFN achieves less visual diversity than the texture network, regardless of the fact that MEFN samples do have larger entropy. We hypothesize that this curious behavior is due to the optimization achieving a local optimum in which the brick boundaries and dark brick locations are not diverse but the entropy within each brick is large. It should also be noted that among the experiments that we ran, this was the only negative example that we got, and that slightly modifying the hyperparameters caused the issue to disappear.

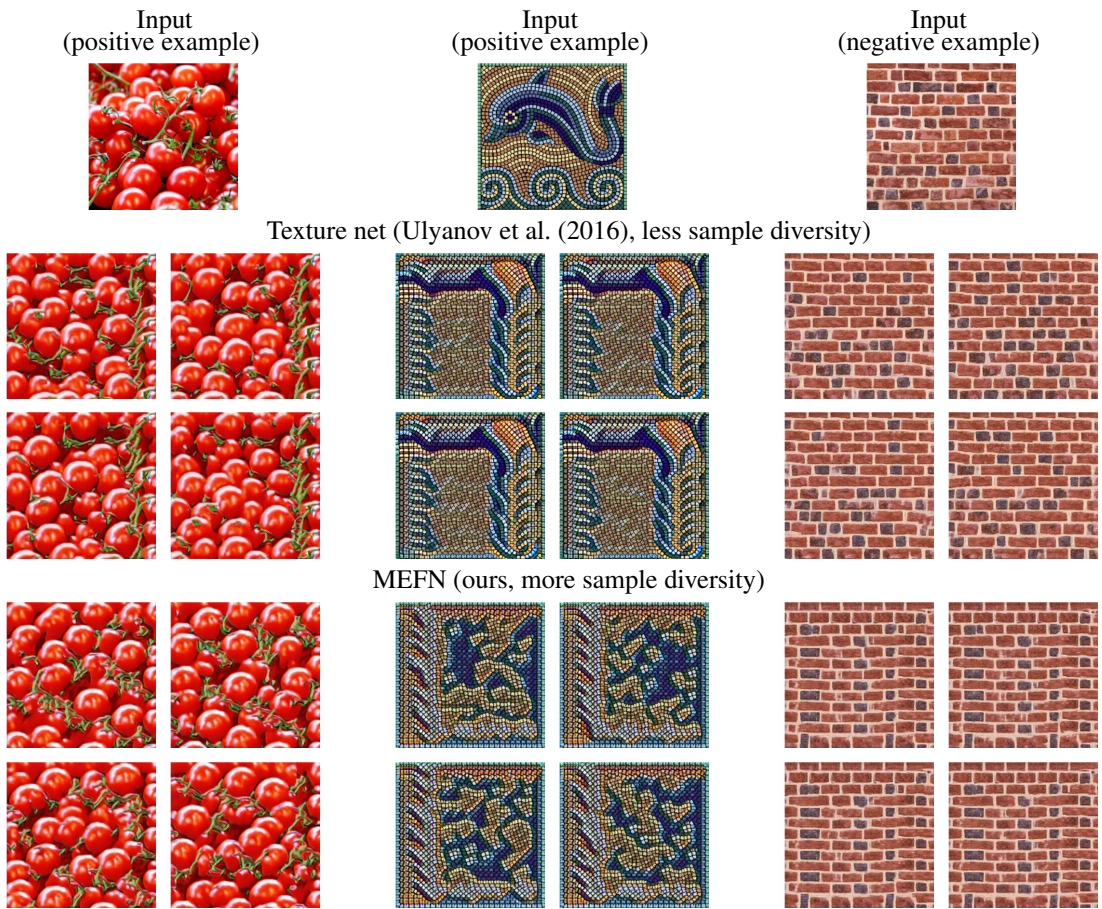

Figure 5: MEFN and texture network samples.

