# Peer review of "Maximum Entropy Flow Networks"

_ICLR 2017 — accepted_

[Official Review · AnonReviewer1 · rating 9 · confidence 5 · 16 Dec 2016]
**Flexible Maximum Entropy Models**
originality 2 · clarity 5 · impact 3 · substance 2 · recommendation (unofficial) 2

Much existing deep learning literature focuses on likelihood based models. However maximum entropy approaches are an equally valid modelling scenario, where information is given in terms of constraints rather than data. That there is limited work in flexible maximum entropy neural models is surprising, but  is due to the fact that optimizing a maximum entropy model requires (a) establishing the effect of the constraints on some distribution, and formulating the entropy of that complex distribution. There is no unbiased estimator of entropy from samples alone, and so an explicit model for the density is needed. This challenge limits approaches. The authors have identified that invertible neural models provide a powerful class of models for solving the maximum entropy network problem, and this paper goes on to establish this approach. The contributions of this paper are (a) recognising that, because normalising flows provide an explicit model for the density, they can be used to provide unbiased estimators for the entropy (b) that the resulting Lagrangian can be implemented as a relaxation of a augmented Lagrangian (c) establishing the practical issues in doing the augmented Lagrangian optimization. As far as the reviewer is aware this work is novel – this approach is natural and sensible, and is demonstrated on an number of models where clear evaluation can be done. Enough experiments have been done to establish this is an appropriate method, though not that it is entirely necessary – it would be good to have an example where the benefits of the flexible flow transformation were much clearer. Further discussion of the computational and scaling aspects would be valuable. I am guessing this approach is probably appropriate for model learning, but less appropriate for inferential settings where a known model is then conditioned on particular instance based constraints? Some discussion of appropriate use cases would be good. The issue of match to the theory via the regularity conditions has been brought up, but it is clear that this can be described well, and exceeds most of the theoretical discussions that occur regarding the numerical methods in other papers in this field.

Quality: Good sound paper providing a novel basis for flexible maximum entropy models.
Clarity: Good.
Originality: Refreshing.
Significance: Significant in model development terms. Whether it will be an oft-used method is not clear at this stage.

Minor issues

Please label all equations. Others might wish to refer to them even if you don’t.
Top of page 4: algorithm 1→ Algorithm 1.
The update for c to overcome stability appears slightly opaque and is mildly worrying.  I assume there are still residual stability issues? Can you comment on why this solves all the problems?
The issue of the support of p is glossed over a little. Is the support in 5 an additional condition on the support of p? If so, that seems hard to encode, and indeed does not turn up in (6). I guess for a Gaussian p0 and invertible unbounded transformations, if the support happens to be R^d, then this is trivial, but for more general settings this seems to be an issue that you have not dealt? Indeed in your Dirichlet example, you explicitly map to the required support, but for more complex constraints this may be non trivial to do with invertible models with known Jacobian? It would be nice to include this in the more general treatment rather than just relegating it to the specific example.

Overall I am very pleased to see someone tackling this question with a very natural approach.

[Official Review · AnonReviewer2 · rating 6 · confidence 4 · 16 Dec 2016 (modified: 22 Jan 2017)]
**OK method, but lack of strong evaluation on real-world problems and lack of significant methodological contributions**

The authors propose a new approach for estimating maximum entropy distributions
subject to expectation constraints. Their approach is based on using
normalizing flow networks to non-linearly transform samples from a tractable
density function using invertible transformations. This allows access to the
density of the resulting distribution. The parameters of the normalizing flow
network are learned by maximizing a stochastic estimate of the entropy
obtained by sampling and evaluating the log-density on the obtained samples.
This stochastic optimization problem includes constraints on expectations with
respect to samples from the normalizing flow network. These constraints are
approximated in practice by sampling and are therefore stochastic. The
optimization problem is solved by using the augmented Lagrangian method. The
proposed method is validated on a toy problem with a Dirichlet distribution and
on a financial problem involving the estimation of price changes from option
price data.

Quality:

The paper seems to be technically sound. My only concern would the the approach
followed to apply the augmented Lagrangian method when the objective and the
constraints are stochastic. The authors propose their own solution to this
problem, based on a hypothesis test, but I think it is likely that this has
already been addressed before in the literature. It would be good if the
authors could comment on this.

The experiments performed show that the proposed approach can outperform Gibbs
sampling from the exact optimal distribution or at least be equivalent, with
the advantage of having a closed form solution for the density.

I am concern about the difficulty of he problems considered.
The Dirichlet distributions are relatively smooth and the distribution in the
financial problem is one-dimensional (in this case you can use numerical
methods to compute the normalization constant and plot the exact density).
They seem to be very easy and do not show how the method would perform in more
challenging settings: high-dimensions, more complicated non-linear constraints,
etc...

Clarity:

The paper is clearly written and easy to follow.

Originality:

The proposed method is not very original since it is based on applying an
existing technique (normalizing flow networks) to a specific problem: that of
finding a maximum entropy distribution. The methodological contributions are
almost non-existing. One could only mention the combination of the normalizing
flow networks with the augmented Lagrangian method. 

Significance:

The results seem to be significant in the sense that the authors are able to
find densities of maximum entropy distributions, something which did not seem
to be possible before. However, it is not clearly how useful this can be in
practice. The problem that they address with real-world data (financial data)
could have been solved as well by using 1-dimensional quadrature. The authors
should consider more challenging problems which have a clear practical
interest.

Minor comments:

More details should be given about how the plot in the bottom right of Figure 2 has been obtained.

"a Dirichlet whose KL to the true p∗ is small": what do you mean by this? Can you give more details on how you choose that Dirichlet?

I changed updated my review score after having a look at the last version of the paper submitted by the authors, which includes new experiments.

[Author Response · Gabriel Loaiza-Ganem · 20 Dec 2016]
**Reviews answers**

We thank our reviewers for their input. We did some minor updates to the manuscript. Here are our answers to the points raised in the reviews:

1. Augmented Lagrangian with stochastic objective and constraints, and how our use of a hypothesis test helps (AnonReviewer2 and AnonReviewer1):

The augmented Lagrangian method transforms a constrained optimization problem into a sequence of unconstrained optimization problems (similar to a log barrier or other interior point methods).  Thus, as long as we are confident in convergence for each unconstrained problem, the overall convergence of the augmented Lagrangian is inherited. This fact remains the case regardless of the underlying optimizer, be it a standard noiseless gradient method, or (as is in this case) an SGD method.  This explanation certainly is somewhat informal, but our experience empirically is that there are no incremental problems with a series of SGD unconstrained optimizations save the minor issue addressed below.

A potential minor issue with the augmented Lagrangian method is that if c is too large, although theoretically not an issue, in practice this will make the unconstrained problems ill-conditioned thus making it hard to solve them numerically. The update rules for c are designed to address this issue. In our experiments we found that sometimes the random nature of our constraint estimates caused c to be updated when it shouldn't (this was the case for the Dirichlet experiments, not the financial ones). Our hypothesis test solution aims to have a more conservative updating rule in order to avoid this issue, and it performed well in practice. It should also be noted that when the number of samples used for the hypothesis test goes to infinity, the hypothesis test becomes the regular update rule (i.e. noiseless).

To the best of our knowledge, there are few sources in the literature on constrained stochastic optimization. There is research addressing constrained optimization when we only have access to a single sample ([1]), but this is not our case as we have access to as many samples as desired. There is a paper ([2]) in which it is proved that alternating a gradient step with a lambda update will work (similar to what we do: alternating optimizing with many gradient steps and performing a lambda update), but the issue of updating c is not touched there, it is just assumed that c is large enough. This does however, at least partially, justify our augmented Lagrangian approach.


2. Higher-dimensional Experiments (AnonReviewer2 and AnonReviewer1):

First, we note that evaluating the baseline in higher-dimensional settings is not trivial: recovering the Gibbs distribution is computationally intractable.  That said, we are currently working on applying our method to generate texture images. We believe this might be a particularly interesting higher dimensional application, as the networks trained to accomplish this task are usually trained to match some statistics, having no guarantee that the generated images are "fair" samples of the objective texture instead of simply very similar images to the input one.  To that end, another paper currently under review at ICLR, "What does it take to generate natural textures?" by Ivan Ustyuzhaninov, Wieland Brendel, Leon Gatys and Matthias Bethge (

[Official Review · AnonReviewer4 · rating 6 · confidence 4 · 03 Jan 2017]
**Application of normalizing flows to max-ent**
recommendation (unofficial) 4

This paper applies the idea of normalizing flows (NFs), which allows us to build complex densities with tractable likelihoods, to maximum entropy constrained optimization.

The paper is clearly written and is easy to follow.

Novelty is a weak factor in this paper. The main contributions come from (1) applying previous work on NFs to the problem of MaxEnt estimation and (2) addressing some of the optimization issues resulting from stochastic approximations to E[||T||] in combination with the annealing of Lagrange multipliers.
Applying the NFs to MaxEnt is in itself not very novel as a framework. For instance, one could obtain a loss equivalent to the main loss in eq. (6) by minimizing the KLD between KL[p_{\phi};f], where f is the unormalized likelihood f \propto exp \sum_k( - \lambda_k T - c_k ||T_k||^2  ). This type of derivation is typical in all previous works using NFs for variational inference.
A few experiments on more complex data would strengthen the paper's results. The two experiments provided show good results but both of them are toy problems.


Minor point:

Although intuitive, it would be good to have a short discussion of step 8 of algorithm 1 as well.

[Author Response · Gabriel Loaiza-Ganem · 11 Jan 2017]
**Last Review Answer**

We thank AnonReviewer4 for the input.  We respond to your points individually:

First, you make an interesting connection with the KLD, but that is not quite applicable here: our formulation has c_k(E T_k)^2 instead of c_k E(T_k ^2), which can't be cast straightforwardly in a VI framework. Having this term is useful because:

+ The true optimum has log density sum eta_i T_i (eta is as in equation 2), with no c_i T_i^2 term. If we didn't have the last term, we would be minimizing the KLD to a log density sum lambda_i T_i, which would only recover the maximum entropy distribution if lambda actually corresponded to eta. Like we mentioned on the paper, computing eta is computationally intractable.

+ Having this term enables the use of the augmented Lagrangian method, giving us the theoretical guarantees discussed on our previous answers, which ensures that we recover the maximum entropy distribution (up to regularity and expressivity of the model class, as discussed).

Second, you commented about novelty in the paper.  We respectfully disagree.  For example, the above KLD example is in fact different than suggested, so the MEFN is not the typical variational inference setup.  Further, we think the introduction of the ME problem is novel to the deep learning community, and conversely, we have introduced deep learning to the ME problem, a literature in which there is substantial need for estimation and sampling techniques.  Mechanically, constrained optimization is rarely used in the deep learning community, and we introduce new steps there.  Finally, per your and others’ requests, we think our new expanded results section adds novelty and state of the art performance to those problem domains (see below).

Third, you asked for more complex data.  Thank you.  We have significantly addressed this concern in the new version of the paper (see updated pdf).  Section 4.3 now details a texture synthesis problem, and demonstrates that the MEFN performs consistently with state of the art in terms of matching texture-net moment constraints, all while outperforming substantially in terms of sample variability.  As the purpose of this class of implicit generative models is to generate a diverse sample set of plausible images, having the ME component in this problem domain is critical and adds to the novelty and impact of our work. 

To your point about step 8 of our algorithm, it simply corresponds to the stochastic version of the usual augmented Lagrangian method, namely lambda_{k+1} = lambda_k + c_k * T(phi_k). This basically corresponds to a gradient step in lambda with step size c_k. The step size is justified by the theoretical derivation of the augmented Lagrangian method, which can be read in detail in Bertsekas (2014).

[Author Response · Gabriel Loaiza-Ganem · 11 Jan 2017]
**Significant Paper Update**

We have significantly revised the paper in response to reviewer and other comments.  Major changes include:

+ Most significantly, we added new experiments applying MEFN to texture synthesis (Section 4.3).  Current state-of-the-art (Ulyanov et al 2016) defines a complicated texture loss and then learns a network that generates images with a small texture loss, with excellent results.  One potential downside in this construction, common to many state of the art deep learning approaches, is that there is no objective encouraging sample diversity; this implies the extreme pathological case where the distribution is simply a point mass on the training image.   We apply the real NVP (Dinh et al 2016), a recently proposed flow tailored to image applications, as our network structure in an MEFN framework.  The MEFN provides images with qualitative textures at least as good as existing state of the art (and this is quantitatively supported), and critically the MEFN produces samples with significantly more diversity (as quantified in a variety of ways).  This experiment highlights the difficulty of obtaining the max entropy distribution in high dimensions and with complicated constraints, and our MEFN handles this setting well and improves the current state-of-the-art in texture synthesis.  As such we feel that this enhanced results section speaks to any concerns about experiments and further increases the novelty and utility of the MEFN framework.

+ In line with the point about sample diversity, we also augmented our Dirichlet experiments of Section 4.1 to show the danger of not considering the entropy term: there moment matching is achieved per the objective, but sample diversity is significantly lost.  This result foreshadows the more important implication of this fact seen in Section 4.3.

+ We have thoroughly revised the text to respond to other reviewer comments and to clarify the contributions and novelties introduced in this work, which we and the reviewers felt were not adequately emphasized before.

[Author Response · Gabriel Loaiza-Ganem · 19 Jan 2017]
**Friendly reminder**

Due to the upcoming end of the rebuttal period this Friday, we kindly remind our reviewers of the recent major update that we did to the paper, and discussions that we added. We feel that we addressed the main concerns raised by the reviewers and would be glad if the reviewers could have a look and share any feedback. Thank you very much.

[Author Response · Gabriel Loaiza-Ganem · 28 Apr 2017]
**Revision**

Added more data as shown on the conference poster, per audience request.

[Final Decision · Program Chairs · 06 Feb 2017 (modified: 14 Feb 2017)]
**ICLR committee final decision**

This paper is an interesting application to maximum entropy problems, which are not widely considered in deep learning. The reviewers have concerns over the novelty of this method and the ultimate applicability and scope of these methods. But this method should be of interest, especially in connecting deep learning to the wider community on information theory and should make an interesting contribution to this year's proceedings.